# In Vitro and In Vivo Regulation of *SRD5A* mRNA Expression of Supercritical Carbon Dioxide Extract from *Asparagus racemosus* Willd. Root as Anti-Sebum and Pore-Minimizing Active Ingredients

**DOI:** 10.3390/molecules27051535

**Published:** 2022-02-24

**Authors:** Warintorn Ruksiriwanich, Chiranan Khantham, Pichchapa Linsaenkart, Tanakarn Chaitep, Pensak Jantrawut, Chuda Chittasupho, Pornchai Rachtanapun, Kittisak Jantanasakulwong, Yuthana Phimolsiripol, Sarana Rose Sommano, Chaiwat Arjin, Houda Berrada, Francisco J. Barba, Korawan Sringarm

**Affiliations:** 1Department of Pharmaceutical Sciences, Faculty of Pharmacy, Chiang Mai University, Chiang Mai 50200, Thailand or ckhantham@gmail.com (C.K.); pichchapa_li@cmu.ac.th (P.L.); tanakarn_c@cmu.ac.th (T.C.); pensak.j@cmu.ac.th (P.J.); chuda.c@cmu.ac.th (C.C.); 2Cluster of Research and Development of Pharmaceutical and Natural Products Innovation for Human or Animal, Chiang Mai University, Chiang Mai 50200, Thailand; sarana.s@cmu.ac.th (S.R.S.); korawan.s@cmu.ac.th (K.S.); 3Cluster of Agro Bio-Circular-Green Industry, Faculty of Agro-Industry, Chiang Mai University, Chiang Mai 50100, Thailand; pornchai.r@cmu.ac.th (P.R.); or kittisak.jan@cmu.ac.th (K.J.); yuthana.p@cmu.ac.th (Y.P.); 4Faculty of Agro-Industry, Chiang Mai University, Chiang Mai 50100, Thailand; 5Department of Animal and Aquatic Sciences, Faculty of Agriculture, Chiang Mai University, Chiang Mai 50200, Thailand; chaiwat_arjin@cmu.ac.th; 6Department of Preventive Medicine and Public Health, Food Science, Toxicology and Forensic Medicine, Faculty of Pharmacy, University of Valencia, 46100 Valencia, Spain; houda.berrada@uv.es (H.B.); francisco.barba@uv.es (F.J.B.)

**Keywords:** anti-sebum efficacy, *Asparagus racemosus*, facial-pore-minimizing efficacy, facial sebum production, 5-alpha reductase enzymes, human volunteer, oily skin, polyphenols, *SRD5A*, supercritical carbon dioxide fluid extraction

## Abstract

Oily skin from overactive sebaceous glands affects self-confidence and personality. There is report of an association between steroid 5-alpha reductase gene (*SRD5A*) expression and facial sebum production. There is no study of the effect of *Asparagus racemosus* Willd. root extract on the regulation of *SRD5A* mRNA expression and anti-sebum efficacy. This study extracted *A. racemosus* using the supercritical carbon dioxide fluid technique with ethanol and investigated its biological compounds and activities. The *A. racemosus* root extract had a high content of polyphenolic compounds, including quercetin, naringenin, and *p*-coumaric acid, and DPPH scavenging activity comparable to that of the standard L-ascorbic acid. *A. racemosus* root extract showed not only a significant reduction in *SRD5A1* and *SRD5A2* mRNA expression by about 45.45% and 90.86%, respectively, but also a reduction in the in vivo anti-sebum efficacy in male volunteers, with significantly superior percentage changes in facial sebum production and a reduction in the percentages of pore area after 15 and 30 days of treatment. It can be concluded that *A. racemosus* root extract with a high content of polyphenol compounds, great antioxidant effects, promising downregulation of *SRD5A1* and *SRD5A2*, and predominant facial sebum reduction and pore-minimizing efficacy could be a candidate for an anti-sebum and pore-minimizing active ingredient to serve in functional cosmetic applications.

## 1. Introduction

Excessively oily facial skin with greasy, shiny, and large pores causes unpleasant feelings and affects personality. The over-production of facial oil from overactive sebaceous glands is the major cause of oily facial skin. Moreover, this skin type is prone to acne and seborrheic dermatitis [1]. Oily or greasy skin is associated with larger pores [2]. Sebum is secreted from sebaceous glands, which are composed of sebaceous follicles [3]. Sebaceous glands are distributed around the face, chest, and upper back [4]. Sebocytes are the predominating cells around sebaceous glands, and they express dihydrotestosterone (DHT) receptors, which influence sebaceous glands’ activity in sebum secretion [3,4]. Androgens mainly stimulate sebum production. DHT, a potent androgen, binds to the androgen receptor in the cytoplasm and then translocates to the nucleus for androgen-regulated gene interaction. This signal orderly regulates the proliferation of sebaceous glands, sebum production, and inflammatory cascades [5]. Furthermore, steroid 5-alpha reductase enzymes play a crucial role in converting testosterone into the more potent DHT. The inhibition of steroid 5-alpha reductase gene (*SRD5A)* expression can decrease the amount of DHT in cells, leading to facial sebum control. Finasteride, a 5-alpha reductase type 2 inhibitor, was approved by the USFDA for benign prostatic hypertrophy and androgenetic alopecia treatment [6]. Dutasteride, a dual type 1 and 2 5-alpha reductase inhibitor, was approved for benign prostatic hypertrophy treatment. Both inhibitors are widely used for baldness treatment. However, oral finasteride and dutasteride can inhibit 5-alpha reductase enzymes all over the body, resulting in serious systemic side effects, such as impotence, a decreasing in sexual ability, and ejaculation disorder. Various medicinal plants that can inhibit *SRD5A* gene expression, such as saw palmetto [7], rice bran, bamboo [8], hemp [9], curcumin [10], and pao pareira [11], have been introduced. Previous studies reported an association between *SRD5A* and facial sebum production [12]. Some cations, especially zinc, have been reported to reduce sebum production in vivo and have been used to treat acne. In vitro assays have indicated that zinc specifically inhibits 5-alpha reductase enzymes. This inhibition may be mediated by both non-competitive inhibition of testosterone binding to 5-alpha reductase enzymes and reduced formation of the NADPH co-factor [13]. Moreover, some medicinal plant extracts, such as saw palmetto, sesame seeds, and argan oil, which have high polyphenolic compounds, have been introduced for facial sebum control [1].

The prescribed medicines for oily skin treatment include topical retinoids, oral isotretinoin, spironolactone, and contraceptives. Most of them have serious side effects, including dry skin, and isotretinoin is contraindicated in pregnancy [4]. Recently, natural products, such as green tea, guava, and L-carnitine topical, have been widely used in anti-seborrheic cosmeceutical products [14,15,16]. *Asparagus racemosus* Willd., known as Shatavari, is a well-known medicinal plant in the Genus *Asparagus*, Family *Asparagaceae*. In Ayurveda, *A. racemosus* is recommended as an immunoadjuvant and lactation booster [17]. Moreover, most studies have reported various biological activities of *A. racemosus* extract, such as immunomodulatory [18], galactagogue [19], antibacterial [20,21], antifungal [22], and antioxidant [23,24] properties. Additionally, *A. racemosus* extract has profound antibacterial and antifungal activities against *Staphylococcus aureus* [20,21] and *Malassezia furfur* [22], which are related to inflammatory skin diseases, such as seborrheic dermatitis [25,26]. Additionally, previous studies have reported a tremendous antioxidant effect of the extract, which resulted in gastric ulcer healing and potential hepatoprotection [23,27]. Interestingly, Himplasia^TM^ containing Shatavari has been prescribed for benign prostatic hyperplasia, and it was claimed to be an *SRD5A1* inhibitor [28]. Several studies have reported that steroidal saponins, isoflavones, flavonoids, and quercetin are the most notable compounds identified in *A. racemosus* root extract [17,29,30]. Steroidal saponins mainly play a crucial part in oxytocin-induced contraction blockage [31,32,33].

Only a few studies have investigated the biological activities and applications of the root extract of *A. racemosus* for functional cosmetic applications [20,22,34], while there are no studies on its anti-sebum efficacy. Therefore, this study aimed to identify the bioactive compounds of *A. racemosus* root extract and determine its effects on antioxidant potential and *SRD5A* gene regulation. The in vivo efficacy and safety evaluation of the effect of *A. racemosus* extract solution on sebum control and pore-minimizing capabilities allowed us to investigate the potential of *A. racemosus* extract for cosmetic anti-sebum applications.

## 2. Results and Discussion

### 2.1. Extraction Yield and Active Compounds of Asparagus racemosus Willd. Root Extract

The extraction yield from supercritical carbon dioxide (scCO_2_) extraction was 1.08% (weight of extract/dry weight of plant material). scCO_2_ was used with ethanol as a cosolvent in this study since ethanol can extract more polar compounds, such as phenolic compounds [35], than scCO_2_ alone. The results of the polyphenol and flavonoid contents in *A. racemosus* are presented in Table 1. Six phenolic compounds were found. Quercetin (3.403 ± 0.412 mg/g extract) predominated, followed by naringenin (0.746 ± 0.027 mg/g extract), and *p*-coumaric acid (0.721 ± 0.010 mg/g extract). Caffeic acid (0.197 ± 0.018 mg/g extract), naringin (0.021 ± 0.007 mg/g extract), and rosmarinic acid (0.012 ± 0.006 mg/g extract) were the minor components of the root extract of *A. racemosus.* In addition, previous reports showed the presence of polyphenolic compounds from the aerial parts of *A. racemosus* [29,36]. Interestingly, steroidal saponins, polysaccharides, glycosides, sterols, and triterpenoids were also found in the roots of *A. racemosus* [24,27,29]. Novel steroidal saponins, such as shatavarins I-V, shatavarosides A and B, asparinins, asparosides, curillins, and curillosides, have previously been isolated from a root of Shatavari by acetonitrile or methanolic extraction [31,32,33], and they could be found in the *A. racemosus* extract from our study.

### 2.2. Antioxidant Effects of Asparagus racemosus Willd. Root Extract Solution

The antioxidant activity was investigated by DPPH, ABTS radical scavenging methods, and ferrous ion (Fe^2+^) chelating assay. The assay for the capability to scavenge stable free radicals was based on measuring the reduction of DPPH radicals in purple to a reduced form of DPPH in yellow [37]. Subsequently, ABTS radical cations in the blue chromophore disappeared after the acceptance of hydrogen radicals from antioxidant compounds [38]. In addition to the metal chelating activity, an evaluation of chelator could reduce the color intensity of red Fe^2+^-ferrozine complexes [39]. The antioxidant values observed in the *A. racemosus* solution were SC_50_ of 0.502 ± 0.275 mg/mL, SC_50_ of 5.319 ± 0.327 mg/mL, and MC_50_ of 1.591 ± 0.175 mg/mL against DPPH, ABTS radicals, and ferrous ion, respectively (Table 2). These antioxidant capacities of extract via ABTS and chelation methods are lower than Trolox and EDTA standards (*p* < 0.05). Nevertheless, DPPH radical scavenging activity of the extract showed a comparable effect to that of L-ascorbic acid. The antioxidant property of the extract might be due to quercetin, which contains three rings and five hydroxyl groups [40]. The chemical structure of quercetin can trap metal ions, as well as reactive oxygen species [41]. Moreover, topical formulations containing gallic acid and quercetin can reduce skin sebum secretion [42]. It can be concluded that the antioxidant abilities of *A. racemosus* are associated with polyphenol contents.

### 2.3. Effects of Asparagus racemosus Willd. Root Extract on 5-Alpha Reductase Isoenzymes

Abnormal sebum production and sebaceous gland enlargement because of excessive androgen levels cause androgen-related skin disorders, such as acne [43]. Testosterone is altered to the most active form of androgen, DHT, by the 5-alpha reductase enzyme [44]. Higher concentrations of the 5-alpha reductase enzyme in sebaceous glands are associated with acne-prone skin [45]. *SRD5A1*, expressed as 5-alpha reductase enzyme type 1, can be found in sweat glands, hair follicles, skin, and especially in facial sebaceous glands [46,47]. Furthermore, *SRD5A2*, expressed as 5-alpha reductase enzyme type 2, is found in hair follicles and sebaceous gland ducts [47,48]. However, *SRD5A3* is predominantly expressed in prostate cancer cells [49]. Consequently, the effects of *SRD5A1* and *SRD5A2* suppression may possibly involve the diminishing of excessive sebum production.

In this study, anti-sebum activity was determined by the expression of *SRD5A* isoenzyme assay using the DU-145 cell line. The non-toxic concentration of the *A. racemosus* extract solution was 10 mg/mL equivalent to the *A. racemosus* extract of 0.5 mg/mL. In contrast, the maximum non-toxic concentration of the standard references (finasteride and dutasteride) was 0.1 mg/mL. These concentrations, which showed more than 90% cell viability [50], were chosen for 5-alpha reductase isoenzyme activity analysis. In this experiment, finasteride, a selective 5-alpha reductase enzyme type 2 inhibitor, and dutasteride, a dual inhibitor of 5-alpha reductase enzymes type 1 and 2 [51], were selected for standard 5-alpha reductase enzyme inhibitor treatment at a concentration of 0.1 mg/mL. The effects of *A. racemosus* root extract on the mRNA expression of *SRD5A* isoenzymes compared with the standard 5-alpha reductase inhibitors (finasteride and dutasteride) were demonstrated based on the percentages of *SRD5A* expression (Figure 1). The inhibitory effects of finasteride on *SRD5A1, SRD5A2,* and *SRD5A3* levels were 68.15 ± 0.10%, 85.27 ± 1.02%, and 62.50 ± 2.18% of control, respectively. In addition, for dutasteride, *SRD5A1*, *SRD5A2,* and *SRD5A3* mRNAs were downregulated to 66.19 ± 0.25%, 80.08 ± 0.16%, and 62.91 ± 0.09% of control, respectively. *A. racemosus* root extract significantly decreased *SRD5A1* and *SRD5A2* mRNA expression by 45.45 ± 0.86% and 90.86 ± 0.06% of control, respectively (*p <* 0.05). Zinc has been reported to reduce sebum production in vivo and to inhibit *SRD5A* via non-competitive inhibition of testosterone binding to *SRD5A* and the decreasing formation of the NADPH cofactor [13]. Many herbal extracts could alleviate androgen-stimulated sebum production from potent *SRD5A* inhibitor activity [44,52,53]. Moreover, saw palmetto, sesame seeds, and argan oil, which have high contents of polyphenolic compounds, have been introduced for use in facial sebum control [1]. The high content of polyphenols, quercetin, naringenin, and coumaric acid in the *A. racemosus* extract solution could exhibit an important role in the reduction of *SRD5A1* and *SRD5A2* mRNA expression in cells. 

### 2.4. Effect on Sebum Level and Pore Area in Volunteers

All volunteers thoroughly accomplished the study protocol. This study compared 5% *w*/*v* of *A. racemosus* root extract (*A. racemosus* extract solution) with control (5% of propylene glycol). The facial sebum contents were measured by Sebumeter^®^, which is a widely accepted technique [54]. The results of the percent changes in facial sebum production as anti-sebum efficacy values are illustrated in Figure 2. The *A. racemosus* extract solution group showed a higher anti-sebum effect than that of the control group in both groups (male and female) at both time intervals (15 and 30 days of treatment). In particular, after 15 days of application, male volunteers in the *A. racemosus* extract solution group showed significantly higher anti-sebum efficacy values on the left cheek area than those of the control group (*p <* 0.05), as shown in Figure 2a. The *A. racemosus* extract solution group demonstrated better anti-sebum efficacy on the left and right cheek areas than that at the forehead area in both groups (male and female). Figure 2b shows that the anti-sebum efficacy of the *A. racemosus* extract solution group in female volunteers was higher than that in the control group on the right and left cheeks at both time intervals (15 and 30 days of treatment). This might be due to the sensitivity of the sebaceous glands to 5-alpha reductase inhibitors in each specific area. The higher sensitivity of the sebaceous glands on the cheeks to the *A. racemosus* extract than the sensitivity of those on the forehead leads to a lower amount of sebum production on both cheeks. A significant anti-sebum effect was only found in male volunteers. A lower amount of facial sebaceous glands and smaller sebaceous units are found in females [48,55], which may have led to the lower sensitivity of the androgen receptor to anti-androgen compounds and the lower anti-sebum efficacy of female volunteers in this study. A previous study reported that androgen levels in females are lower than in males [56]. The proliferation of sebocytes is stimulated by androgen, resulting in a larger sebaceous gland and a higher level of sebum production in males than in females [56]. These reasons could explain the better response of males over females to the *A. racemosus* root extract.

The percentages of pore areas (large and fine) were evaluated based on facial visualization from VisioFace^®^ CSI software. The percentages of pore area reduction after 15 and 30 days of application were calculated, and they are displayed in Figure 3 and Figure 4. In the male volunteers, the *A. racemosus* extract solution group showed a significant reduction in both large and fine pore areas on the forehead compared to the control group (*p <* 0.05) at a period of 15 days in Figure 3a, b, respectively. A similar significant reduction in large pore areas on the forehead was found (*p <* 0.05) at a period of 30 days, as shown in Figure 3a. Interestingly, a superior pore size reduction was demonstrated on the forehead area compared to the cheek area in both large and fine pore areas. Figure 4 shows an example of the before/after 30 days result of the *A. racemosus* extract solution treatment. This male volunteer had an initial large pore area of 3.20 ± 0.28% and a fine pore area of 2.48 ± 0.13%, and after 30 days of treatment, he had a large pore area of 2.15 ± 0.10% (large pore area reduction of 32.78%) and a fine pore area of 2.28 ± 0.03% (fine pore area reduction of 7.18%). The lower pore area reduction was shown in the female volunteers, complying with our previous results. Moreover, the treatment time interval to provide a significant anti-sebum effect was 15 days of treatment in the male volunteer group and 30 days of treatment in the female volunteer group. We can conclude that gender affected the anti-sebum activity of the *A. racemosus* extract solution. In the previous study, excessive sebum levels, age, and the male gender affected facial pore size enlargement [56]. Facial pore sizes further play an important role in releasing sebum from sebaceous glands [52]. Androgen is predominantly exerted to the proliferation of sebocytes and sebum production [56,57]. Yoon et al., reported that polyphenolic compounds in medicinal plants played an important role in decreasing sebum production via the anti-lipogenic effect [58]. Previous in vivo studies of *Camellia sinensis* (green tea), *Quercus acutissima* bark, and *Galdieria sulphuraria* extracts, which effectively inhibit 5-alpha reductase activity, showed a reduction in sebum secretion. These effects might be from polyphenol compounds, such as pentagalloyl glucose, eugeniin, and gallic acid [42,57,59].

Moreover, the activity of the 5-alpha reductase enzyme and the production of testosterone and DHT in oily skin are upregulated. As a result, 5-alpha reductase is the major enzyme involved in the conversion of testosterone to the more potent DHT [13,46]. The topical application of 5% of the *A. racemosus* root extract solution with a high content of polyphenol compounds showed a superior anti-sebum potential and pore reduction effect in male volunteers. These results may imply that *A. racemosus* root extract could downregulate *SRD5A1* and *SRD5A2* expression, thereby decreasing sebum production and providing the pore-minimizing ability of the in vivo test.

For a subjective assessment, the results of the volunteers’ satisfaction of product characteristics, skin tolerance, and efficacy are shown in Figure 5. All volunteers from the *A. racemosus* extract groups stated that the extract solution had great potential to decease facial grease, minimize pore sizes, and provide smoother skin. However, the unique odor of the *A. racemosus* extract was not satisfactory.

## 3. Materials and Methods

### 3.1. Chemicals and Reagents

Finasteride and dutasteride were purchased from Wuhan W&Z Biotech (Wuhan, China). Trypsin, Roswell Park Memorial Institute medium (RPMI-1640), fetal bovine serum (FBS), penicillin, and streptomycin were obtained from Gibco (Thermo Fisher Scientific, Waltham, MA, USA). 2,2-Diphenyl-1-picrylhydrazyl (DPPH), 2,2′-azino-bis (ethylbenzthiazoline-6-sulfonic acid (ABTS), 3-(2-yyridyl)-5,6-diphenyl-1,2,4-triazine-4′,4′′-disulfonic acid sodium salt (ferrozine), iron (II) chloride tetrahydrate (FeCl_2_ · 4H_2_O), L-ascorbic acid, Trolox, ethylenediaminetetraacetic acid (EDTA), and sulforhodamine B (SRB) were obtained from Sigma Chemical (St. Louis, MO, USA). Agarose gel, Tris base, and 50X Tris/acetic acid/EDTA (TAE) were obtained from Bio-Rad Laboratories (Hercules, CA, USA). Ethanol, acetic acid, trichloroacetic acid, and other chemical substances were obtained from RCI Labscan (Bangkok, Thailand). All other chemical substances were of analytical grade.

### 3.2. Preparation of Sample

The dried root of Shatavari (*Asparagus racemosus* Willd.) was provided by Bangkok Lab & Cosmetic Co., Ltd. (Ratchaburi, Thailand). Plant material was prepared by supercritical carbon dioxide (scCO_2_) fluid extraction following a patent-pending process. Dried and crushed *A. racemosus* roots (1 kg) were placed into the extractor (Guangzhou Gongcheng Digital Science Technology, Guangzhou, China) operating at 50 °C and 30 MPa for 1 h, with 1 L of 95% (*v*/*v*) ethanol as a cosolvent. The extract solution was concentrated using a rotary evaporator (Hei-VAP value, Heidolph, Schwabach, Germany) at temperature up to 50 °C. The *A. racemosus* extract was stored at 4 °C until further use for analysis. The *A. racemosus* extract was diluted in the propylene glycol at the concentration of 5% *w*/*v* (*A. racemosus* extract solution) for the antioxidant, 5-alpha reductase isoenzymes, and in vivo efficacy test.

### 3.3. Analysis of Phenolic Compounds in Asparagus racemosus Willd. Root Extract by Liquid Chromatography–Mass Spectrometry (LC-MS)

The Agilent 1260 Infinity II series (Agilent Tech., Santa Clara, CA, USA) was connected to an electrospray ionization (ESI) quadrupole mass spectrometry 6130 equipped with a degasser, binary pump, column oven, and thermostatted autosampler, and analysis was performed according to our previously reported method [60]. The reversed-phase C18 analytical column (250 × 4.6 mm, 4.6 mm, 5 µm (Restek Corporation, Bellefonte, PA, USA) was used for separation. A column was kept at 30 °C, and the flow rate was 0.5 mL/min. Then, 5 µL of the sample volume (10 mg/mL of sample was dissolved in ethanol and filtrated through a 0.45 µm syringe filter) was injected. Gradient elution was performed using 5% formic acid as solvent A and acetonitrile: H_2_O: formic acid (85:10:5) as solvent B. A linear gradient was performed as follows: 0–8 min, 80% A; 8–24 min, decreased A to 25%; 24–28 min, 25% A; 28–34 min, increased A to 70%; 34–36 min, increased A to 80%; 36–45 min, 80% A. Mass analysis of compounds was performed using negative ion monitoring (SIM). Nitrogen was used as a drying gas, and the program was performed as follows: flow rate, 12 L/min; drying gas temperature, 350 °C; nebulizer pressure, 60 psi; capillary voltage, 3000 V; fragmentor voltage, 70 V; and full-scan spectra from 100 to 1200 m/z with 250 ms/spectrum. OpenLab software (Agilent Tech., Santa Clara, CA, USA) was used for spectra analysis. The list of phenolic compounds, the limit of detection, and the limit of quantification are provided in Appendix A.

### 3.4. Antioxidant Activities Analysis

#### 3.4.1. DPPH Radical Scavenging Activity

The DPPH assay was performed using the previous protocol with slight modifications [50]. The result was compared to that of L-ascorbic acid. The sample was diluted by distillated water in the range of 0.01–10 mg/mL. Briefly, 50 µL of the sample was reacted with 50 µL of 0.1 mM DPPH solution, which was freshly prepared in ethanol. The mixtures were incubated at room temperature in the dark for 30 min. The absorbance was measured at 517 nm against a blank (ethanol), using a microplate reader (EZ Read 2000, Biochrom, Holliston, MA, USA). The percentages of the DPPH radical scavenging activity were calculated by Equation (1), where Abs_Control_ is the absorbance of the DPPH solution, and Abs_Sample_ is the absorbance of the DPPH radicals that reacted with the sample:(1)$$\mathrm{DPPH}\;\mathrm{radical}\;\mathrm{scavenging}\;\mathrm{activity}\;\left(\%\right)=\frac{{\mathrm{Abs}}_{\mathrm{Control}\;}-{\;\mathrm{Abs}}_{\mathrm{Sample}}}{{\mathrm{Abs}}_{\mathrm{Control}}}\times 100.\;$$

The concentration providing 50% scavenging activity (SC_50_) (mg/mL) was obtained from the linear relationship between the concentration of the samples and the percentages of the DPPH radical scavenging activity.

#### 3.4.2. ABTS Radical Scavenging Activity

The ABTS assay was evaluated according to the previous method with some modifications [50]. The assay was based on ABTS radical scavenging ability in comparison to that of Trolox and L-ascorbic acid in the range of 0.01–10 mg/mL. The ABTS stock solutions were prepared by placing 7 mM of ABTS radical solution in distilled water reacted with 2.45 mM of potassium persulfate solution, and they were stored at room temperature in the dark for 12–16 h. Then, the ABTS working solution was diluted to obtain an absorbance of 0.7–0.9 at 734 nm with distilled water using the microplate reader (EZ Read 2000, Biochrom, Holliston, MA, USA). The sample (25 µL) was reacted with 200 µL of ABTS solution for 10 min. The percentages of the ABTS radical scavenging activity were calculated by Equation (2), where Abs_Control_ is the absorbance of the ABTS solution, and Abs_Sample_ is the absorbance of the ABTS radicals that reacted with the sample:(2)$$\mathrm{ABTS}\;\mathrm{radical}\;\mathrm{scavenging}\;\mathrm{activity}\;\left(\%\right)=\frac{{\mathrm{Abs}}_{\mathrm{Control}}-{\;\mathrm{Abs}}_{\mathrm{Sample}}}{{\mathrm{Abs}}_{\mathrm{Control}}}\times 100.\;$$

The concentration providing 50% scavenging activity (MC_50_) (mg/mL) was obtained from the linear relationship between the different concentrations of the samples and the percentages of the ABTS radical scavenging activity.

#### 3.4.3. Metal Chelating Activity

Iron chelating activity was analyzed by a previously reported method [60]. Briefly, 50 µL of 5 mM ferrozine was mixed with 50 µL of FeCl_2_ · 4H_2_O solution. The sample was diluted with distillated water to use concentrations ranging from 0.01 to 10 mg/mL. Then, 100 µL of the sample reacted with the solution of ferrozine-Fe^2+^ complex for 30 min at room temperature. EDTA was used as a standard. The absorbance of ferrozine-Fe^2+^ complex was measured at 562 nm using the microplate reader (EZ Read 2000, Biochrom, Holliston, USA). The percentages of metal chelating activity were calculated by Equation (3), where Abs_Control_ is the absorbance of the ferrozine-Fe^2+^ complex, and Abs_Sample_ is the absorbance of the ferrozine-Fe^2+^ complex reacted with the sample:(3)$$\mathrm{Metal}\;\mathrm{chelating}\;\mathrm{activity}\;\left(\%\right)=\frac{{\mathrm{Abs}}_{\mathrm{Control}}-{\;\mathrm{Abs}}_{\mathrm{Sample}}}{{\mathrm{Abs}}_{\mathrm{Control}}}\times 100.\;$$

The concentration providing 50% chelating activity (MC_50_) (mg/mL) was obtained from the linear relationship between the different concentrations of the samples and the ferrous scavenging activity.

### 3.5. 5-Alpha Reductase Isoenzyme Activity Analysis

#### 3.5.1. Cell Culture

DU-145 human prostate cancer cells were obtained from the American Type Culture Collection (Rockville, MD, USA). DU-145 cells were cultured in RPMI-1640 and supplemented with 10% (*v*/*v*) FBS, 100 µg/mL of streptomycin, and 100 unit/mL of penicillin. The cells were maintained in a temperature-controlled (37 °C) and humidified incubator containing 5% CO_2_ (CCL-050B-8, Esco^®^, Singapore).

#### 3.5.2. Determination of Cell Viability

The non-cytotoxic concentration of the samples on DU-145 cells was determined by sulforhodamine B (SRB) assay, as previously described [50]. Human prostate cancer cells (1 × 10^5^ cells/mL) were cultured in 96-well plates for 24 h at 37 °C and 5% CO_2_. The cells were tested with the samples in the range of 0.0001–10 mg/mL for another 24 h. After incubation, the adherent cells were fixed in situ by 50% trichloroacetic acid and dyed with 0.04% SRB solution, which was prepared in 1% acetic acid. The bound dye was solubilized, and the absorbance was measured at 515 nm using the microplate reader (EZ Read 2000, Biochrom, Holliston, MA, USA). The percentages of cell viability were calculated by Equation (4), where Abs_Control_ is the absorbance of cells cultured with the medium without supplementation, Abs_Blank_ is the absorbance of cells treated with the solvent, and Abs_Sample_ is the absorbance of cells treated with the sample:(4)$$\mathrm{Cell}\;\mathrm{viability}\;\left(\%\right)=\frac{{\mathrm{Abs}}_{\mathrm{Sample}}-{\;\mathrm{Abs}}_{\mathrm{Blank}}}{{\mathrm{Abs}}_{\mathrm{Control}}-{\;\mathrm{Abs}}_{\mathrm{Blank}}}\;\times 100$$

#### 3.5.3. RNA Extraction and Semi-Quantitative Reverse Transcription Polymerase Chain Reaction (RT-PCR) Analysis

The regulation of *SRD5A* gene expression was tested as previously described [61]. Initially, DU-145 cells were treated with the samples for 24 h. Total RNA was extracted using the NucleoSpin^®^ RNA isolation kit (Macherey-Nagel, Duren, Germany). The extracted RNA was quantified by Qubit 4 fluorometer (Invitrogen, Carlsbad, CA, USA) and Qubit™ RNA HS Assay Kit (Invitrogen). The complementary DNA (cDNA) was generated from the RT-PCR Quick Master Mix (Toyobo, Osaka, Japan) according to the instructions of the manufacturer. Gene-specific primers were used as follows: *SRD5A1*, F: 5′-AGCCATTGTGCAGTGTATGC-3′ and R: 5′-AGCCTCCCCTTGGTATTTTG-3′; *SRD5A2*, F: 5′-TGAATACCCTGATGGGTGG-3′ and R: 5′-CAAGCCACCTTGTGGAATC-3′ ; *SRD5A3*, F: 5′-TCCTTCTTTGCCCAAACATC-3′ and R: 5′-CTGATGCTCTCCCTTTACGC-3′ ; the reference gene (glyceraldehyde 3-phosphate dehydrogenase (GAPDH)), F: 5′-GGAAGGTGAAGGTCGGAGTC-3′ and R: 5′-CTCAGCCTTGACGGTGCCATG-3′. The expression of *SRD5A* was calculated relative to *GAPDH* expression. The percentages of *SRD5A* suppression were calculated by Equation (5), where RE_Control_ is the expression of *SRD5A* genes on untreated cells relative to GAPDH value, and RE_Sample_ is the expression of *SRD5A* genes on treated cells relative to GAPDH value:(5)$$SRD5A\;\mathrm{suppression}\;\left(\%\right)=\frac{{\mathrm{RE}}_{\mathrm{Control}}-{\;\mathrm{RE}}_{\mathrm{Sample}\;}}{{\mathrm{RE}}_{\mathrm{Control}}}\times 100.\;$$

### 3.6. Efficacy Evaluation

#### 3.6.1. Study Population

The study was approved by the Ethics Committee of Chiang Mai University (approval number 002/2564/F), and informed consent was obtained from volunteers. Thai healthy volunteers, 20 males and 22 females, who were aged 18–60 years [62] and had mean sebum values of more than 220 µg/cm^2^ on forehead and 180 µg/cm^2^ on cheeks, measured using Sebumeter^®^ (SM815; Courage + Khazaka Electronic GmbH, Cologne, Germany) according to the guidelines of the manufacturer, were enrolled in this randomized, double-blinded, placebo-controlled study. The subjects were allocated to either the control group (propylene glycol) (*n* = 21) or the active group (*A. racemosus* root extract) (*n* = 21). Exclusion criteria included pregnancy, breastfeeding, irritation to cosmetic products, use of topical corticosteroids and/or vitamin A derivative and/or oil control products, and current dermatological pathologies.

#### 3.6.2. Measurement of Anti-Sebum Efficacy and Pore Area Reduction

The efficacy evaluation was determined by following the modified method [16]. The skin tests were carried out in a room of 20 ± 1 °C and 40–60% relative humidity. The subjects were allowed to rest in the room for 30 min before randomization to apply 1 pump of the 5% of propylene glycol or 5% of *A. racemosus* root extract twice a day. The facial sebum contents and full-face photography at baseline and at 15 and 30 days of treatment period were performed using Sebumeter^®^ and VisioFace^®^ 1000D Camera, respectively, with inbuilt Complete Skin Investigation (CSI) analysis software (Courage + Khazaka Electronic GmbH, Cologne, Germany). A subject withdrew from the study after skin irritation at the application site. The facial sebum values of forehead, left cheek, and right cheek were acquired from Sebumeter^®^ measurement. The percentages of anti-sebum efficacy were calculated by Equation (6), where St is the sebum values at the time interval, and S0 is the sebum values at baseline:(6)$$\mathrm{Anti}-\mathrm{sebum}\;\mathrm{efficacy}\;\left(\%\right)=\frac{\mathrm{St}}{S0}\times 100.\;$$

The percentages of pore area at baseline and at different times of treatment period were obtained from VisioFace^®^ reports. The percentages of pore area reduction were calculated by Equation (7), where Pt is the percentages of the pore area at the time interval, and P0 is the percentages of the pore area at baseline:(7)$$\mathrm{Pore}\;\mathrm{area}\;\mathrm{reduction}\;\left(\%\right)=100-\left(\;\frac{\mathrm{Pt}}{P0}\times 100\right)$$

#### 3.6.3. Self-Assessment

A subjective evaluation questionnaire regarding the product characteristics (odor, color, and stickiness), tolerance, and efficacy was evaluated by grading scores on a 1 to 5 scale (1 is the minimum and 5 is the maximum score of satisfaction) after 30 days of twice-daily application.

### 3.7. Statistical Analysis

All experiments were performed in at least triplicate for each test. Data are expressed as means ± standard deviation (SD). One-way ANOVA, followed by LSD’s post hoc test, was used to analyze the significant differences using SPSS 23.0 software (SPSS Inc., Chicago, IL, USA). The efficacy determinations were analyzed with an independent *t* test. A value of *p <* 0.05 is considered statistically significant.

## 4. Conclusions

*Asparagus racemosus* Willd. root was extracted using the supercritical CO_2_ technique with ethanol as a cosolvent. The *A. racemosus* root extract had a high content of polyphenolic compounds, including quercetin, naringenin, and coumaric acid, providing DPPH radical scavenging activity comparable to that of standard L-ascorbic acid. The anti-sebum efficacy of the *A. racemosus* extract solution was determined via in vitro and in vivo assays. An in vitro assay of the mRNA expression of *SRD5A* isoenzymes (types 1–3) was conducted and compared with that of the standard 5-alpha reductase enzyme inhibitors (finasteride and dutasteride). The 5-alpha reductase enzyme inhibitors showed an association with the decrease in facial sebum production. Not only did *A. racemosus* root extract show a significant reduction in *SRD5A1* and *SRD5A2* mRNA expression by 45.45 ± 0.86% and 90.86 ± 0.06%, but it also showed a reduction in in vivo anti-sebum efficacy in male volunteers in the percentage changes in facial sebum production and percentages of pore area reduction after 15 and 30 days of treatment. This anti-sebum effect was not shown in the female volunteers; i.e., there was a lower percentage of changes in facial sebum production and lower percentages of pore area reduction after treatment in the female group. However, the number of volunteers should be increased in future studies. From the results of this study, it can be concluded that *A. racemosus* root extract, with its high content of polyphenol compounds, promising antioxidant effects, downregulation of *SRD5A1* and *SRD5A2*, and predominant facial sebum reduction and facial pore minimizing efficacy, could be a candidate as an anti-sebum active ingredient to serve in functional cosmetic applications.

## Figures and Tables

**Figure 1 molecules-27-01535-f001:**
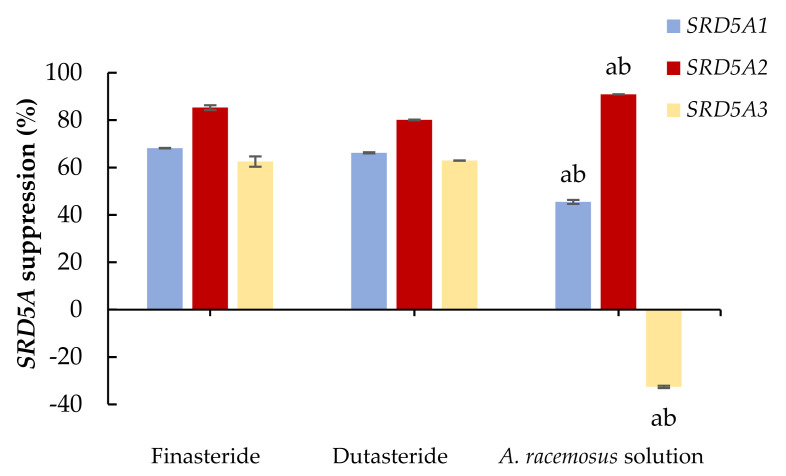
The effects of 0.1 mg/mL of finasteride and dutasteride, and 0.5 mg/mL of *A. racemosus* Willd. root extract solution on 5-alpha reductase isoenzymes expression (*SRD5A*) in human prostate cancer cells. The percentage of *SRD5A* suppression was compared with the control. Statistical significance in comparison to finasteride and dutasteride is indicated as “a” and “b”, respectively (*p <* 0.05).

**Figure 2 molecules-27-01535-f002:**
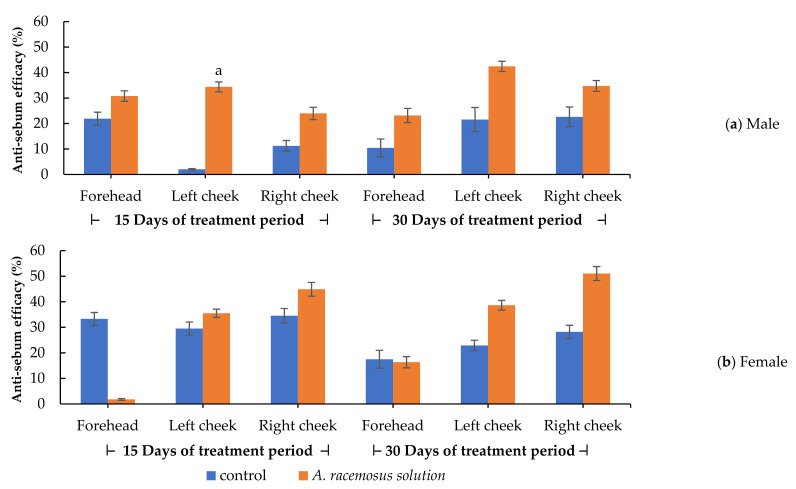
Anti-sebum efficacy of the control (5% of propylene glycol) and the 5% of *Asparagus racemosus* Willd. root extract solution groups on forehead, left cheek, and right cheek. (**a**) Anti-sebum efficacy on forehead, left cheek, and right cheek in male volunteers after 15 days and 30 days of treatment periods. (**b**) Anti-sebum efficacy on forehead, left cheek, and right cheek in female volunteers after 15 days and 30 days of treatment periods. Statistical significance in comparison to the control is indicated as “a” (*p <* 0.05).

**Figure 3 molecules-27-01535-f003:**
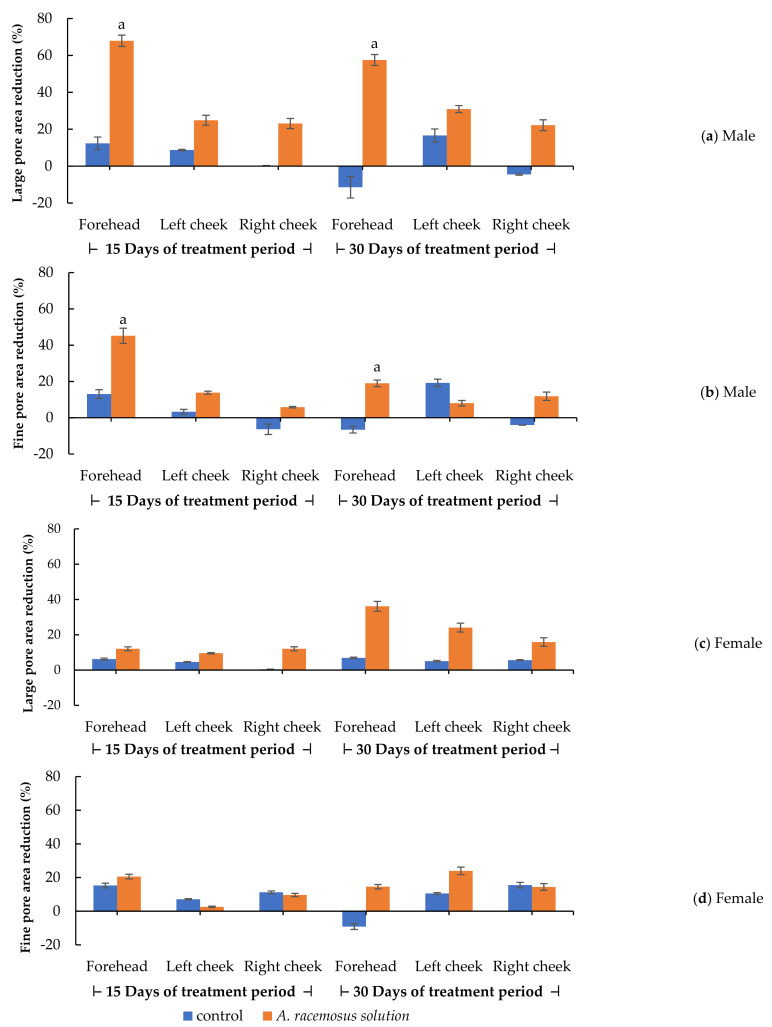
Pore area reduction of the control (5% of propylene glycol) and the 5% of *Asparagus racemosus* Willd. root extract solution groups on forehead, left cheek, and right cheek. (**a**) Large and (**b**) fine pore area reduction after 15 days and 30 days of treatment periods in male volunteers. (**c**) Large and (**d**) fine pore area reduction after 15 days and 30 days of treatment periods in female volunteers. Statistical significance in comparison to the control is indicated as “a” (*p <* 0.05).

**Figure 4 molecules-27-01535-f004:**
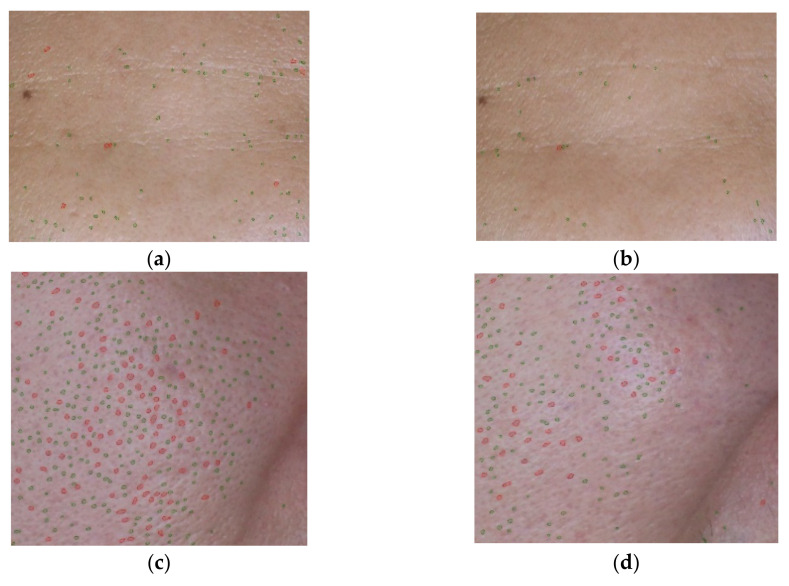
An example of pore area visualization from the *A. racemosus* Willd. root extract solution treatment using VisioFace^®^ CSI software (Köln, Germany) (**a**) before and (**b**) after 30 days of the extract treatment on forehead; (**c**) before and (**d**) after 30 days of the extract treatment on right cheek. Red spots = large pores; green spots = fine pores.

**Figure 5 molecules-27-01535-f005:**
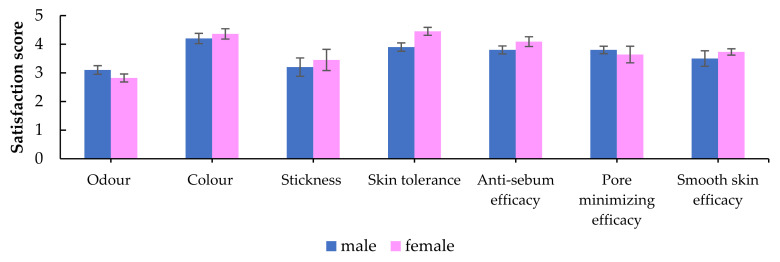
Volunteer satisfaction of the characteristics, tolerance, and efficacy of *A. racemosus* Willd. root extract solution (male *n* = 10, female *n* = 12).

**Table 1 molecules-27-01535-t001:** Polyphenol compositions of *Asparagus racemosus* root extract.

Compositions (mg/g Extract)	
Quercetin	3.403 ± 0.412
Naringenin	0.746 ± 0.027
*p*-Coumaric acid	0.721 ± 0.010
Caffeic acid	0.197 ± 0.018
Naringin	0.021 ± 0.007
Rosmarinic acid	0.012 ± 0.006

Each value is expressed as mean ± SD (*n* = 3).

**Table 2 molecules-27-01535-t002:** Antioxidant activities of *Asparagus racemosus* Willd. root extract solution.

Antioxidant Activities	DPPH Radical Scavenging Activity (SC_50_, mg/mL)	ABTS Radical Scavenging Activity (SC_50_, mg/mL)	Fe^2+^ Chelating Activity (MC_50_, mg/mL)
*A. racemosus* Willd. root extract solution	0.502 ± 0.275	5.319 ± 0.327 ^a^	1.591 ± 0.175 ^a^
L-ascorbic acid	0.154 ± 0.014	0.067 ± 0.006	Nd
Trolox	Nd	0.092 ± 0.003	Nd
EDTA	Nd	Nd	0.063 ± 0.004

Each value is expressed as mean ± SD (*n* = 3); ethylenediaminetetraacetic acid (EDTA); Nd = not determined; SC_50_ = the concentration providing 50% scavenging activity (mg/mL); MC_50_ = the concentration providing 50% chelating activity (mg/mL). A statistical significance in comparison to the control indicated as “a” (*p <* 0.05).

## Data Availability

The data presented in this study are available on request from the corresponding author.

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
