# Peer review of "In Vitro and In Vivo Regulation of *SRD5A* mRNA Expression of Supercritical Carbon Dioxide Extract from *Asparagus racemosus* Willd. Root as Anti-Sebum and Pore-Minimizing Active Ingredients"

_molecules, 2022, doi:10.3390/molecules27051535_

Round 1

Reviewer 1 Report

The authors of In Vitro and In Vivo Regulation of SRD5A mRNA Expression of Asparagus racemosus Willd. Root Extract by Supercritical Carbon Dioxide as Anti-Sebum and Pore-Minimizing Active Ingredient showed the effects of this extract in vivo in male volunteers. Which is the main and valuable result.

As they suggest it would be greatly beneficial for the future study to increase the numbers of volunteers.

Also the discussion on the absence of the effects in female volunteers should be improved.

There are a lot of simple grammatical typos and errors - but there are several sentences that  are way too confusing not to be improved or revised as per readers convenience.

Quality of presentation is the major drawback of this paper and should be significantly improved.

Also the percentages are much more grateful when expressing the results (Figure 1.) than folds - less confusing.

All the comments are marked in yellow in adobe acrobat .pdf file attached.

Author Response

Answers to the Comments of the Reviewer 1

Manuscript ID:                               molecules-1559025

Title:                                                  In Vitro and In Vivo Regulation of SRD5A mRNA Expression of Supercritical Carbon Dioxide Extract from Asparagus racemosus Willd. Root as Anti-Sebum and Pore-Minimizing Active Ingredients

Authors:                                                     Warintorn et al.

Response to comments:

REVIEWER 1

  1. The authors of In Vitro and In Vivo Regulation of SRD5A mRNA Expression of Asparagus racemosus Willd. Root Extract by Supercritical Carbon Dioxide as Anti-Sebum and Pore-Minimizing Active Ingredient showed the effects of this extract in vivo in male volunteers. Which is the main and valuable result.

Response: The main and valuable results of this study are the A. racemosus root extract from scCO2 not only showed the comparable DPPH radical scavenging activity to the standard L-ascorbic acid but also suppressed the mRNA expression of SRD5A, contributing to the reduction of facial sebum production.

Moreover, the in vivo study found that facial sebum reduction and the facial pore minimizing efficacy of A. racemosus root extract were clearly seen significantly in male volunteers and slightly observed in females. These valuable results would support the utilization of the extract for its functional cosmetic application.

  1. As they suggest it would be greatly beneficial for the future study to increase the numbers of volunteers.

Response: Thank you for your comment. In conclusions, the sentence “In our further study, the in vivo assay should be increased the sample size in the experiment.” has been reconstructed to “However, the numbers of volunteers should be increased for the future study.”, on lines 452–453, page 13.

  1. Also, the discussion on the absence of the effects in female volunteers should be improved.

Response: The discussion has been described on lines 188-212, page 5: “All volunteers thoroughly accomplished the study protocol. This study compared 5% w/v of A. racemosus root extract (A. racemosus extract solution) with control (5 % of propylene glycol). The facial sebum contents were measured by Sebumeter®, which is a widely accepted technique [54]. The results of percent changes in facial sebum production as anti-sebum efficacy values were illustrated in Figure 2. The A. racemosus extract solution showed a higher anti-sebum effect than the control group in both groups (male and female) at both time intervals (15 and 30 days of treatment). Especially after 15 days of application, male volunteers in the A. racemosus extract solution group showed significantly higher anti-sebum efficacy values on the left cheek area than the control group (p < 0.05) shown in Figure 2a. The A. racemosus extract solution group demonstrated better anti-sebum efficacy at the left and right cheek area than that at the forehead area in both groups (male and female). Figure 2b showed the anti-sebum efficacy of A. racemosus extract solution group in female volunteers were higher than those in control group at the right and left cheeks at both time intervals (15 and 30 days of treatment). It might be due to the sensitivity to 5-alpha reductase inhibitors of sebaceous glands in each specific area. The higher sensitivity to A. racemosus extract of sebaceous glands at cheeks than those at the forehead leading to the lower amount of sebum production at both cheeks. Significant anti-sebum effect was found only in male volunteers. Since a lower amount of facial sebaceous glands and smaller sebaceous units were shown in females [48,55] leading to the lower sensitivity to anti-androgen compounds at the androgen receptor and the lower anti-sebum efficacy of female volunteers in this study. Previous study reported that androgen levels in females are lower than in males [56]. The proliferation of sebocytes is stimulated by androgen, resulting a larger sebaceous gland and higher level of sebum production in males than females [56]. These reasons explain the better response of A. racemosus root extract of males over females.”.

References:

  1. Zouboulis, C.C.; Degitz, K. Androgen action on human skin–from basic research to clinical significance. Exp Dermatol 2004, 13, 5-10.
  2. Sugawara, T.; Nakagawa, N.; Shimizu, N.; Hirai, N.; Saijo, Y.; Sakai, S. Gender‐and age‐related differences in facial sebaceous glands in Asian skin, as observed by non‐invasive analysis using three‐dimensional ultrasound microscopy. Skin Res Technol 2019, 25, 347-354.

  1. There are a lot of simple grammatical typos and errors - but there are several sentences that are way too confusing not to be improved or revised as per readers convenience.

Response: We are sorry for the mistake. The revised manuscript has been thoroughly checked and corrected via Grammarly (premium) program using UK spelling.

  1. Quality of presentation is the major drawback of this paper and should be significantly improved.

Response: Figures 2 and 3 have been adjusted to clarify the presentation as followings. The caption of figure 2 and 3 has been edited. Please see on pages 6-8.

  1. Also, the percentages are much more grateful when expressing the results (Figure 1.) than folds - less confusing.

Response: The results of mRNA expression have been illustrated “the percentages of SRD5A suppression” instead of “fold change” in Figure 1, page 5.

  1. All the comments are marked in yellow in adobe acrobat. (pdf file attached)

Response: Thank you so much for your valuable suggestion. The revised manuscript has been thoroughly checked and corrected according to the suggestions in highlighted text which are as follows:

  • The phrases “5 alpha reductase” and “5 alpha-reductase” have been changed to “5-alpha reductase” throughout the manuscript.
  • “L-ascorbic acid” and “L-carnitine” have been typed in lower font of “L” on lines 31, 78, 132, 278, 319, 333, 441, and Table 2.
  • The sentence: “It can conclude…” has been altered to “It can be concluded…” on lines 34 and 136.
  • The last paragraph of introduction: “Only a few research investigated the biological activities and applications of root extract of racemosus for functional cosmetic application, especially for the anti-sebum efficacy….” has been edited and added the reference to complement the information to “Only a few studies investigated the biological activities and applications of root extract of A. racemosus for functional cosmetic application [20,22,34], while there are no studies on the anti-sebum efficacy.”. Please see on lines 94-96 of page 2.

References:

  1. Mandal, S.C.; Nandy, A.; Pal, M.; Saha, B. Evaluation of antibacterial activity of Asparagus racemosus Willd. root. Phytother Res 2000, 14, 118-119.
  2. Onlom, C.; Khanthawong, S.; Waranuch, N.; Ingkaninan, K. In vitro anti‐Malassezia activity and potential use in anti‐dandruff formulation of Asparagus racemosus. Int J Cosmet Sci 2014, 36, 74-78.
  3. Rungsanga, T.; Tuntijarukornb, P.; Ingkaninanc, K.; Viyocha, J. Stability and clinical effectiveness of emulsion containing Asparagus racemosus root extract. Sci Asia 2015, 41, 236-245.
  • The sentence: “The supercritical carbon dioxide (scCO2) extraction was achieved the extraction yield 103 of 1.08 % of the dry weight.” has been revised to “The extraction yield from supercritical carbon dioxide (scCO2) extraction was 1.08 % (weight of extract/dry weight of plant material).”. Please see on lines 103-104 of page 3.
  • “Polyphenol compositions” has been converted to “phenolic compounds”, on line 107 of page 3.
  • “Contents” has been changed to “components”, on line 111 of page 3.
  • The table’s names have been settled in the center for Table 1 and Table 2, on pages 3-4, respectively.
  • The sentence: “Subsequently, ABTS radical cations in blue chromophore were disappeared after reaction with antioxidant compounds” has been modified to “Subsequently, ABTS radical cations in blue chromophore were disappeared after acceptance of hydrogen radicals from antioxidant compounds.”. Please see on lines 124-126 of page 3.
  • The sentence: “The antioxidant property of the extract might be due to quercetin, which contained three rings and five hydroxyl groups [40].” has been cited incorrect reference. We are sorry about this; the new reference has been added on line 134 of page 3.

Reference:

  1. Moalin, M.; Van Strijdonck, G.P.; Beckers, M.; Hagemen, G.J.; Borm, P.J.; Bast, A.; Haenen, G.R. A planar conformation and the hydroxyl groups in the B and C rings play a pivotal role in the antioxidant capacity of quercetin and quercetin derivatives. Molecules 2011, 16, 9636-9650.
  • “Which were illustrated in” has been deleted, on line 130 of page 3
  • “Expression” has been turned into “effects”, on lines 144 and 164 of page 4
  • “Significant” has been revised to “significantly”, on line 195 of page 5.
  • The terms “xx %” and “xx °C” have been inserted one space between the numbers and the symbols throughout the manuscript.
  • “Follows” has been changed to “following” on line 287 of page 9.
  • The phrase: “thermostat autosampler was…” has been reconstructed to “thermostatted autosampler and analysis was…”. Please see on line 300 of page 9.
  • “Dying gas” has been corrected to “drying gas”, on lines 310, 311 of page 10.
  • “Spectra” has been edited to “spectra analysis”, on line 313-314 of page 10.
  • “Determined by...” has been converted to “performed using…”, on line 318 of page 10.
  • “Slightly” has been edited to “slight”, on line 318 of page 10.
  • “This solution” has been rewrote to “the solution of ferrozine-Fe2+ complex”, on line 349 of page 10.
  • “…at 37 °C” has been edited to “… (37 °C)”, on line 363 of page 11.
  • “Carbon dioxide” has been shorten to “CO2, on line 438 of page 13.
  • “Predominated” has been removed from line 450 of page 13.
  • “Since” has been changed to “i.e.,”, on line 450 of page 13.
  • The sentence: “In our further study, the in vivo assay should be increased the sample size in the experiment.” has been revised to “However, the numbers of volunteers should be increased for the future study.”, on lines 452-453 of page 13.
  • “The result from…” has been rephrased to “From the results of…”, on line 453 of page 13.
  • “Nazir, Y.; Linsaenkart, P.; Khantham, C.; Chaitep, T.; Jantrawut, P.; Chittasupho, C.; Rachtanapun, P.; Jantanasakulwong, K.; Phimolsiripol, Y.; Sommano, S.R.; et al. High efficiency in vitro wound healing of Dictyophora indusiata extracts via anti-Inflammatory and collagen stimulating (MMP-2 inhibition) mechanisms. J Fungi 2021, 7, 1100.”. This reference has been completed with authors’ names by the EndNote program. Please see in the reference list.

Please see Figures 1, 2, and 3 in the attachment.

Sincerely,

Warintorn et al.

Reviewer 2 Report

The authors accent that up to now there is no study of Asparagus racemosus Willd. root extract on the regulation of SRD5A mRNA expression and anti-sebum efficacy. Therefore, they prepared the extract of A. racemosus root using supercritical carbon dioxide fluid technique with ethanol and investigated its biological compounds and activities. The A. racemosus root extract showed high polyphenol content.

In general, the manuscript describes multi-sided study level, however, shortcomings in the descriptions of results were observed.

Comments:

  1. The title of the article is awkward. As if there are 2 sentences, but the idea is not clear. Sounds like …Regulation of … expression by supercritical carbon dioxide, and CO2 is anti-sebum  active ingredient.
  2. Fold change in Fig. 1 does not show mRNA expression convincingly. Where corresponding mRNA levels in control cells are shown, where standard deviations for the mean values, and what means reference gene bar? To my mind, if comparison is shown as fold changes than something should be the control level taken as 1 or 0.

Why statistical significance is shown only among finasteride and dutasteride but not versus control cell level.  What this comparison tells about the extract effect in the cells?

  1. Line 201 and 202. Where is the evidence for the claim “it did not comply in the female group, which had a lower amount of testosterone, DHT and 5 alpha reductase enzymes.” Have you measured and compared these levels?
  2. Descriptions of images should indicate test substance concentrations and solvent controls. Was the extract studied at only one concentration and how was it found? Is mentioned non-toxic the extract concentration 0,5 mg/ml for cells sufficient argument to further investigate only this one maximal non-toxic  concentration? For example, in antioxidant assays the extract’s efficient concentrations were different.
  3. The description of the methods does not mention the apparatus for measuring OD.
  4. Should volunteers with such an age distribution be selected? Line 386. “healthy volunteers, 20 male and 22 female who aged 18-60 years with the mean sebum ...
  5. The Abstract tells us that the extract shows DDPH scavenging effect, reducing of SRD5A1 and SRD5A2 mRNA expression in human prostate cell line and in vivo anti- sebum efficacy in male volunteers.  Is it enough to conclude that   racemosus root extract possessed a great antioxidant effect and predominant facial sebum reduction and pore-minimizing efficacy to serve for its functional cosmetic application? Maybe then for males cosmetics?

Author Response

Answers to the Comments of the Reviewer 2

Manuscript ID:                               molecules-1559025

Title:                                                  In Vitro and In Vivo Regulation of SRD5A mRNA Expression of Supercritical Carbon Dioxide Extract from Asparagus racemosus Willd. Root as Anti-Sebum and Pore-Minimizing Active Ingredients

Authors:                                                     Warintorn et al.

Response to comments:

REVIEWER 2

The authors accent that up to now there is no study of Asparagus racemosus Willd. root extract on the regulation of SRD5A mRNA expression and anti-sebum efficacy. Therefore, they prepared the extract of A. racemosus root using supercritical carbon dioxide fluid technique with ethanol and investigated its biological compounds and activities. The A. racemosus root extract showed high polyphenol content.

In general, the manuscript describes multi-sided study level, however, shortcomings in the descriptions of results were observed.

  1. The title of the article is awkward. As if there are 2 sentences, but the idea is not clear. Sounds like …Regulation of … expression by supercritical carbon dioxide, and CO2 is anti-sebum active ingredient.

Response: Thank you for your suggestion. We completely agree with your opinion. The manuscript title “In Vitro and In Vivo Regulation of SRD5A mRNA Expression of Asparagus racemosus Willd. Root Extract by Supercritical Carbon Dioxide as Anti-Sebum and Pore-Minimizing Active Ingredient” has been changed to In Vitro and In Vivo Regulation of SRD5A mRNA Expression of Supercritical Carbon Dioxide Extract from Asparagus racemosus Willd. Root as Anti-Sebum and Pore-Minimizing Active Ingredients”.

  1. Fold change in Fig. 1 does not show mRNA expression convincingly. Where corresponding mRNA levels in control cells are shown, where standard deviations for the mean values, and what means reference gene bar? To my mind, if comparison is shown as fold changes than something should be the control level taken as 1 or 0.

Response: Thank you for pointing out our unclear information. The revised manuscript has been reported the percentages of SRD5A suppression instead. Please see in Figure 1, page 5.

  1. Why statistical significance is shown only among finasteride and dutasteride but not versus control cell level. What this comparison tells about the extract effect in the cells?

Response: We are sorry for the mistake. Figure 1 has been edited. The results have been expressed as the percentages of SRD5A suppression.

On page 11, lines 391-395, the expression of SRD5A was calculated relative to the GAPDH expression. Then, the percentages of SRD5A suppression were performed by this equation:

SRD5A suppression (%) =

(RE Control - RE Sample)

× 100

(5)

RE Control

Where, REControl is the expression of SRD5A genes on untreated cells which relative to GAPDH value and RESample is the expression of SRD5A genes on treated cells which relative to GAPDH value. Therefore, the effects of the extract on SRD5A genes would compare to control.

  1. Line 201 and 202. Where is the evidence for the claim “it did not comply in the female group, which had a lower amount of testosterone, DHT and 5 alpha reductase enzymes.” Have you measured and compared these levels?

Response: We have not measured and compared the level of those parameters. However, we have discussed on page 5, lines 205-212; “Since a lower amount of facial sebaceous glands and smaller sebaceous units were shown in females [48,55] leading to the lower sensitivity to anti-androgen compounds at the androgen receptor and the lower anti-sebum efficacy of female volunteers in this study. Previous study reported that androgen levels in females are lower than in males [56]. The proliferation of sebocytes is stimulated by androgen, resulting a larger sebaceous gland and higher level of sebum production in males than females [56]. These reasons ex-plain the better response of A. racemosus root extract of males over females.”.

References:

  1. Zouboulis, C.C.; Degitz, K. Androgen action on human skin–from basic research to clinical significance. Exp Dermatol 2004, 13, 5-10.
  2. Sugawara, T.; Nakagawa, N.; Shimizu, N.; Hirai, N.; Saijo, Y.; Sakai, S. Gender‐and age‐related differences in facial sebaceous glands in Asian skin, as observed by non‐invasive analysis using three‐dimensional ultrasound microscopy. Skin Res Technol 2019, 25, 347-354.

  1. Descriptions of images should indicate test substance concentrations and solvent controls. Was the extract studied at only one concentration and how was it found? Is mentioned non-toxic the extract concentration 0,5 mg/ml for cells sufficient argument to further investigate only this one maximal non-toxic concentration? For example, in antioxidant assays the extract’s efficient concentrations were different.

Response: The cells were tested for the cytotoxicity with the serial 10-fold dilution (in the range of 0.0001 – 10 mg/mL) of A. racemosus solution, finasteride, and dutasteride. The extract at the concentration of 0.5 mg/ml and the reference standards (finasteride and dutasteride) at the concentration of 0.1 mg/ml are the maximum non-toxic concentrations in cell lines. The concentrations of test substances have been added on the description of Figure 2 and 3. On page 4, lines 157-160, the sentences the non-toxic concentration of A. racemosus extract solution was 10 mg/mL equivalent to A. racemosus extract of 0.5 mg/mL. Whereas the maximum non-toxic concentration of standard references (finasteride and dutasteride) was 0.1 mg/mL.” have been revised.

  1. The description of the methods does not mention the apparatus for measuring OD.

Response: The apparatus for measuring OD is the microplate reader EZ Read 2000. Additional information has been inserted as “…using the microplate reader (EZ Read 2000, Biochrom, Holliston, USA)”. Please see on lines 323-324, 337, 351-352 and 372-373 of pages 10-11.

  1. Should volunteers with such an age distribution be selected? Line 386. “Healthy volunteers, 20 male and 22 female who aged 18-60 years with the mean sebum...

Response: We used the similar criteria according to the previous study (reference no. 62) that we can investigate the effect in both young skin and mature skin. On page 12 (lines 339-400), the reference has been inserted in this sentence: “Thai healthy volunteers, 20 males and 22 females who aged 18-60 years [62] with the mean sebum...”.

Reference:

  1. Delsin, S.; Mercurio, D.; Fossa, M.; Maia Campos, P. Clinical efficacy of dermocosmetic formulations containing Spirulina extract on young and mature skin: effects on the skin hydrolipidic barrier and structural properties. Clin Pharmacol Biopharm 2015, 4, 2.

  1. The Abstract tells us that the extract shows DDPH scavenging effect, reducing of SRD5A1 and SRD5A2 mRNA expression in human prostate cell line and in vivo anti- sebum efficacy in male volunteers. Is it enough to conclude that A. racemosus root extract possessed a great antioxidant effect and predominant facial sebum reduction and pore-minimizing efficacy to serve for its functional cosmetic application? Maybe then for males’ cosmetics?

Response: The multimode of biological actions regarding antioxidant and antiandrogen may contribute to the sebum reduction and pore-minimizing effect. The in vivo study found that facial sebum reduction and the facial pore minimizing efficacy were clearly seen significantly in male volunteers and slightly observed in females. However, we could not conclude that the extract is not effective in female due to a few samples size of this study may not give enough statistical power. Further investigations with the larger population are needed. Moreover, we have mentioned this problem on lines 452-453 of page 13; “However, the numbers of volunteers should be increased for the future study.”.

Sincerely,

Warintorn et al.

Reviewer 3 Report

In this study, the authors aimed to determine the metabolite composition of Aparagus racemosus root extract and to evaluate the antioxidant, SRD5A1 and SRD5A2 enzymes inhibitory and anti-sebum activity of this extract.

1) The metabolite composition and the antioxidant activity of Aparagus racemosus root extract have already been investigated, determining many phenolics compounds in this extract and also confirming its antioxidant activity. Comparing the phytochemical and antioxidant results of the manuscript with those of previous literature data, the reviewer couldn’t find any novelties. What are the new results of this work in these topics?

2) Six common phenolics were only identified in the extract; however, many other, relevant compounds are also known in the literature.

3) Amounts of identified compounds are given in the manuscript; however, the method of quantitation, used by the authors, is missing. 

4) The total amount of compounds is rather small as this value is only 5.1 mg/g (i.e., 0,51% of the extract is “known”).

The study analyzing the enzyme inhibitory and anti-sebum activity of extract may be of interest to a more specific journal dealing with dermatology or cosmetology.

In conclusion the manuscript, in its present form doesn’t meet the requirements of the Molecules.

Author Response

Answers to the Comments of the Reviewer 3

Manuscript ID:                               molecules-1559025

Title:                                                  In Vitro and In Vivo Regulation of SRD5A mRNA Expression of Supercritical Carbon Dioxide Extract from Asparagus racemosus Willd. Root as Anti-Sebum and Pore-Minimizing Active Ingredients

Authors:                                                     Warintorn et al.

Response to comments:

REVIEWER 3

In this study, the authors aimed to determine the metabolite composition of Asparagus racemosus root extract and to evaluate the antioxidant, SRD5A1 and SRD5A2 enzymes inhibitory and anti-sebum activity of this extract.

  1. The metabolite composition and the antioxidant activity of Asparagus racemosus root extract have already been investigated, determining many phenolics compounds in this extract and also confirming its antioxidant activity. Comparing the phytochemical and antioxidant results of the manuscript with those of previous literature data, the reviewer couldn’t find any novelties. What are the new results of this work in these topics?

Response: In this study, the main aims are to verify the functional application of A. racemosus root extract and provide the evident support for its cosmetic uses which could fit in the scope of the journal’s special issue “the functional applications of medicinal plants”. This manuscript provides the new perspective of A. racemosus root extract regarding the antioxidant activities, the in vitro regulation of SRD5A expressions, and in vivo evaluation of anti-sebum and pore area reduction.  No earlier study identified those biological activities and in vivo evaluation of the extract in facial sebum control. The phenolic compounds, which have shown interesting bioactivities, were focused and could be used to support the discussion of this study. However, we understand your point that we did not extensively identify other compounds or novel compounds. We believed that this work would contribute to the deeper investigation of other bioactive compounds related to their biological effects in specific pathways in the future.

  1. Six common phenolics were only identified in the extract; however, many other, relevant compounds are also known in the literature.

Response: Twelve phenolic compounds have been investigated and screened. However, only six compounds have been detected in the extracts. We are sorry that we did not mention the list of all compounds. We have provided the detail of compounds and their parameter in Supplementary Materials. On page 10 (lines 314-315), the sentence “The list of phenolic compounds, the limit of detection, and the limit of quantification were in Supplementary Materials (Table S1).” has been added. Moreover, we have discussed other compounds that may be found in the extract on page 3 (lines 111-117).

  1. Amounts of identified compounds are given in the manuscript; however, the method of quantitation, used by the authors, is missing.

Response: The method of liquid chromatography-mass spectrometry (LC-MS) was described on pages 9-10 (lines 296-315). The list of phenolic compounds, the limit of detection, and the limit of quantification have been given in Supplementary Materials (Table S1).

  1. The total amount of compounds is rather small as this value is only 5.1 mg/g (i.e., 0,51% of the extract is “known”).

Response: We understand your concern. However, the main aims of this study are to determine the biological effects of Aparagus racemosus root extract for the cosmetic application. Even the identification of bioactive compounds is our weak point. We have mentioned other compounds that may be found in the extract on page 3 (lines 111-117).

  1. The study analyzing the enzyme inhibitory and anti-sebum activity of extract may be of interest to a more specific journal dealing with dermatology or cosmetology.

Response: Thank you for your kind comment.

  1. In conclusion the manuscript, in its present form doesn’t meet the requirements of the Molecules.

Response: We are very much appreciated your suggestion. All comments we received have been taken to improve the quality of the revised manuscript

Sincerely,

Warintorn et al.

Round 2

Reviewer 1 Report

Dear Authors, I can see that you have taken an effort an corrected much of your paper. You have also recreated the Figures which in my opinion adds to the quality of presentation.

Only thing that remains to be corrected (and it was not included in the first version of this paper) is the Supplementary material.

You state that the linearity range (i.e. calibration curve) is the same (1.56 -100 μg/mL) for all phenolic compounds except for Quercetin (1.56 -50 μg/mL).

However, all your LOD values (μg/mL) are 3 to 8 times higher than your first point of calibration curve (1.56 μg/mL), while LOQ values are roughly 3,03 times higher.

This, however, is impossible, since LOD values should be lower than first point of calibration curve (1.56 μg/mL), while LOQ should be, at best, the same as the first point of calibration curve (1.56 μg/mL). Please re-check these values and correct them accordingly, also, please give reference to the method you used to calculate LOD and LOQ values.

Reviewer 2 Report

The manuscript is significantly improved. I do not have critical comments.

Reviewer 3 Report

The authors answered the reviewer’s questions and completed their manuscript accordingly. Thus, the present manuscript is acceptable for publication.
